# Mechanistic models of PLC/PKC signaling implicate phosphatidic acid as a key amplifier of chemotactic gradient sensing

**Jamie L. Nosbisch**[1], **Anisur Rahman**[2], **Krithika Mohan**[2], **Timothy C. Elston**[3], **James E. Bear**[4], **Jason M. Haugh**[2]*

**1** Biomathematics Graduate Program, North Carolina State University, Raleigh, North Carolina, United States of America, **2** Department of Chemical and Biomolecular Engineering, North Carolina State University, Raleigh, North Carolina, United States of America, **3** Department of Pharmacology, University of North Carolina School of Medicine, Chapel Hill, North Carolina, United States of America, **4** Department of Cell Biology and Physiology, UNC Lineberger Comprehensive Cancer Center, University of North Carolina School of Medicine, Chapel Hill, North Carolina, United States of America

* jason_haugh@ncsu.edu

**Data Availability Statement:** All relevant data are within the paper and its Supporting Information files, and associated Virtual Cell simulations results

## Abstract

Chemotaxis of fibroblasts and other mesenchymal cells is critical for embryonic development and wound healing. Fibroblast chemotaxis directed by a gradient of platelet-derived growth factor (PDGF) requires signaling through the phospholipase C (PLC)/protein kinase C (PKC) pathway. Diacylglycerol (DAG), the lipid product of PLC that activates conventional PKCs, is focally enriched at the up-gradient leading edge of fibroblasts responding to a shallow gradient of PDGF, signifying polarization. To explain the underlying mechanisms, we formulated reaction-diffusion models including as many as three putative feedback loops based on known biochemistry. These include the previously analyzed mechanism of substrate-buffering by myristoylated alanine-rich C kinase substrate (MARCKS) and two newly considered feedback loops involving the lipid, phosphatidic acid (PA). DAG kinases and phospholipase D, the enzymes that produce PA, are identified as key regulators in the models. Paradoxically, increasing DAG kinase activity can enhance the robustness of DAG/active PKC polarization with respect to chemoattractant concentration while decreasing their whole-cell levels. Finally, in simulations of wound invasion, efficient collective migration is achieved with thresholds for chemotaxis matching those of polarization in the reaction-diffusion models. This multi-scale modeling framework offers testable predictions to guide further study of signal transduction and cell behavior that affect mesenchymal chemotaxis.

## Author summary

Cell movement directed by external gradients of chemical composition is critical for immune responses, wound healing, and development. Although theoretical concepts explaining how shallow external gradients might definitively polarize a cell's motility have been offered over the past two decades, mathematical models cast in terms of defined molecules and mechanisms are uncommon in this context. Based on both recent and older

are publicly available as indicated under Materials and Methods.

**Funding:** This work was supported by National Institutes of Health grant U01-EB018816 to JMH. The funders had no role in study design, data collection and analysis, decision to publish, or preparation of the manuscript.

**Competing interests:** The authors have declared that no competing interests exist.

insights from the literature, we offer mechanistic models that are able to explain experimentally observed polarization of signal transduction elicited by shallow attractant gradients. A novel insight of our models is the implicated role of phosphatidic acid, a membrane lipid produced by at least two enzymatic pathways, in two positive feedback loops that amplify signal transduction locally. In separate simulations, we explored the implications of polarization for efficient cell invasion during wound healing. We expected that the ability to polarize in response to shallow gradients would enhance the speed of wound invasion, but an unexpected finding is that this property can promote intermittent polarization throughout the wound.

## Introduction

Chemotaxis, the bias of cell movement towards soluble chemical cues (chemoattractants), is critical for embryonic development, angiogenesis, the immune response, and wound healing in metazoans [1]. Fibroblasts, the cells directly responsible for regenerating wounded tissue, respond chemotactically to platelet-derived growth factor (PDGF) as a cue to invade the wound; this proliferative phase of wound healing typically unfolds over the course of several days [2–5]. Chemoattractant ligands such as PDGF bind to cognate receptors on the cell surface, and in eukaryotic cells this signal is sensed spatially, relying on a gradient of receptor occupancy and activation. The activated receptors interface with a network of intracellular signaling pathways that modulate the dynamics of the cytoskeleton and thus cell motility. This modulation can be achieved by spatially varying the rate of F-actin polymerization or the mechanical influence of Myosin II contractility [6,7]. In fibroblasts exposed to a steady PDGF gradient, many of the prominent signaling pathways that enhance the rate of F-actin polymerization have been found to be dispensable for chemotaxis [8–10], whereas regulation of Myosin IIA by phospholipase C (PLC)/protein kinase (PKC) signaling, a well-studied pathway activated by many receptors, is essential [11]. Another key finding in that study was that diacylglycerol (DAG), the lipid product of PLC that activates most PKC isoforms, is sharply concentrated in the fibroblasts' protrusions (lamellipodia) exposed to the highest concentration of PDGF [11]. Given that the external gradients in such experiments are characteristically shallow (typically, ~ 5% across a cell's length), the PLC/PKC signaling circuit must be locally amplified to explain the observed polarization of DAG production. What are the biochemical and biophysical mechanisms that cause the pathway to polarize?

To partially address this question, a reaction-diffusion model of the PLC/PKC signaling pathway identified phosphorylation of myristoylated alanine-rich C kinase substrate (MARCKS) by PKC, which increases the availability of the PLC substrate (PIP$_2$) [12–15], as a positive feedback loop (PFL); this mechanism is sufficient for polarization in response to an abnormally steep external gradient [16]. By itself, however, the MARCKS feedback was unable to polarize signaling in response to shallow external gradients ($\leq$ 10%), and the system lacked robustness to modest changes in the midpoint concentration of chemoattractant. In this work, we address these issues through formulation of more mechanistic, partial differential equation models of the PLC/PKC pathway. These models consider two additional PFLs supported by literature evidence and thus introduce a key molecular player: phosphatidic acid (PA), a lipid intermediate in the metabolism of DAG.

PA is recognized as a signaling molecule affecting a number of cellular functions including cell growth and proliferation, vesicular trafficking, and cytoskeletal rearrangement [17,18]. In the present models, we include reactions by which PA is produced by phosphorylation of

DAG by DAG kinases or from hydrolysis of phosphatidylcholine by phospholipase D (PLD) [19–22]. Feedback loops incorporating PA were added to the model based on published evidence that: 1) PA binds PLCγ and increases the rate of $PIP_2$ hydrolysis in vitro [23]; and 2) active PKC can enhance the activity of PLD for increased production of PA [24–26]. Model simulations show that the MARCKS feedback mechanism synergizes with these new feedback loops to polarize PLC/PKC signaling in response to shallow gradients of receptor occupancy and over an appreciable range of midpoint occupancy. Subtle asymmetry of the cell geometry can also polarize signaling. Focusing on the molecular details, simulations suggest that DAG kinases, the enzymes responsible for turnover of DAG, exert a critical and surprisingly positive influence on the responsiveness of the circuit. Finally, we applied the predicted receptor occupancy conditions for polarization to a two-state model of cell movement in a hybrid (stochastic/continuum) model of wound invasion. This model predicts a hierarchy of chemotactic waves that drive more efficient collective invasion than a single chemotactic front. With this framework, we can proceed to link signal transduction mechanisms at the molecular level to individual and collective cell movements directed by chemoattractant gradients in tissues.

## Results

### New models of the PLC/PKC pathway based on putative feedback mechanisms indicated in the literature

We formulated a biochemically realistic description of the chemotactic sensing circuit in fibroblasts, retaining the differential $PIP_2$ buffering by MARCKS from the Mohan model [16] and considering two additional PFLs based on published evidence (**Fig 1A**; see also *Materials and Methods* and **S1 Text**). The first, PFL 1, considers the modulation of PLC recruitment by phosphatidic acid (PA), the lipid produced by phosphorylation of DAG by DAG kinases. Using a detergent-phospholipid mixed micelle assay system, it was shown that inclusion of PA enhanced $PIP_2$ hydrolysis catalyzed by either unphosphorylated or tyrosine-phosphorylated PLCγ1, by reducing the apparent $K_m$ for the reaction [23]. The details of how PA affects PLC activity are not completely understood, but the reduction of the apparent $K_m$ is consistent with PA-mediated stabilization of PLCγ1 association with the membrane, akin to the effect of the non-catalytic interaction of PLCδ with $PIP_2$ [27]. Therefore, we modeled the effect of PA as an increased lifetime of the receptor-PLCγ1 complex at the membrane (**Fig 1A**).

PFL 2 considers the effect of active PKC on the activity of phospholipase D1 (PLD1), which produces PA by hydrolyzing the abundant lipid, phosphatidylcholine [19,21]. In murine fibroblasts stimulated with phorbol ester, PKCα interacts with PLD1 and increases the rate of phosphatidylcholine hydrolysis. PA produced by this reaction can be dephosphorylated to yield DAG, and thus PFL 2 exerts an influence on PLC/PKC signaling independent of PFL 1. We model the influence of PLD as a distinct source term for generation of PA, which increases according to a Hill function of the active PKC density at the membrane (**Fig 1A**).

To simplify and generalize the handling of receptor dynamics in this model, and considering that receptor activation is patterned by an external ligand, we assume a steady gradient of occupied/active receptors, $r$, that is linear in the direction of the cell's long axis (on both the top and bottom of the cell) rather than along the cell's contour (**Fig 1B & 1C**). In the expression for $r$, *rfrac* is the average receptor occupancy, expressed as a fraction of a characteristic receptor density of 130 μm$^{-2}$, or $10^5$/cell. The parameter *rsteep* is the relative steepness of the receptor occupancy gradient across the cell; for example, a value of *rsteep* = 0.1 corresponds to a 10% difference between the front and back of the cell.

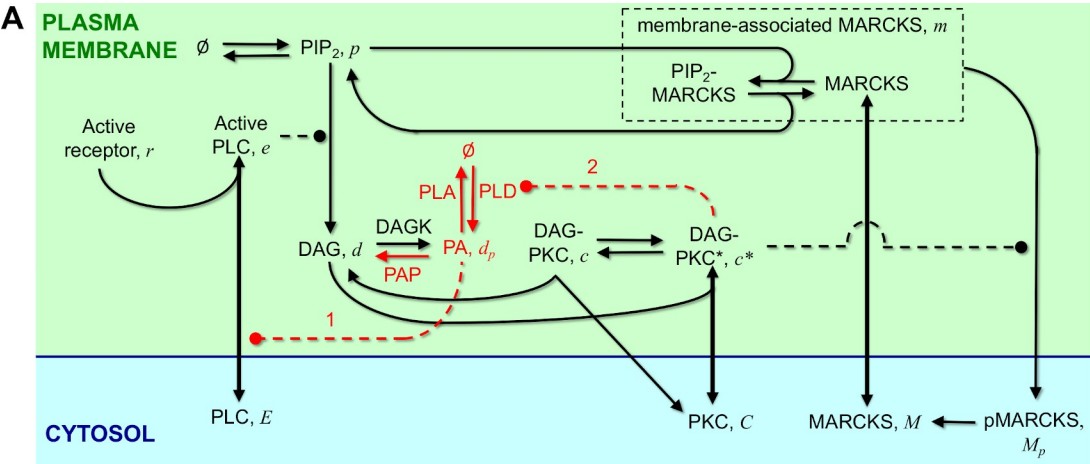

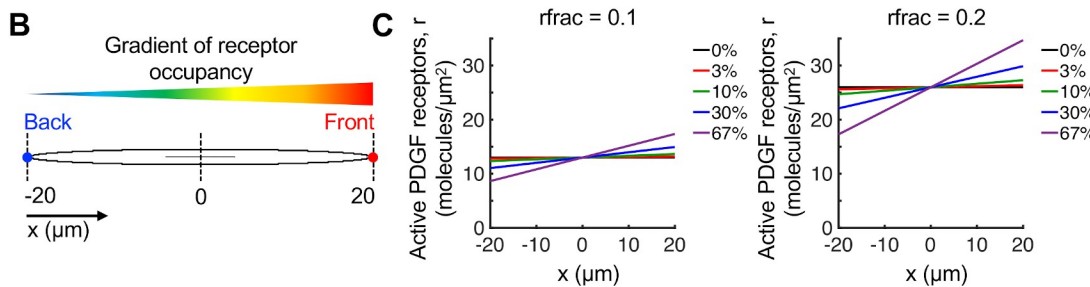

**Fig 1. Model of the PLC/PKC network including phosphatidic acid (PA). (A)** Model schematic depicting the interactions and reactions among signaling proteins and plasma membrane lipids. Dashed lines ending in a filled circle indicate that the species enhances the associated process. The reactions and interactions shown in red are associated with the generation and influence of PA in positive feedback loops (PFLs) labeled (1) and (2). **(B)** Base model geometry and orientation of the receptor occupancy gradient, which is linear in the direction of the cell's long axis, $x$. **(C)** Plots illustrating the linear profile of active receptor density imposed across the 40 μm length of the cell for varying values of *rfrac* (relative midpoint density) and *rsteep* (relative steepness, expressed here as a percentage difference across the cell).

### Stabilization of PLC recruitment by phosphatidic acid (PFL 1), combined with neutralization of MARCKS by PKC, promotes sensitive and robust gradient sensing

Considering the network depicted in **Fig 1A**, with PFL 1 but not PFL 2, we evaluated the ability of receptor occupancy gradients (characterized by midpoint occupancy and % steepness) to polarize DAG and active PKC. For each of five gradient steepness values, ranging from 0% (uniform stimulation) up to 67% (2-fold) difference across the cell, simulations were run varying the value of the midpoint receptor occupancy, *rfrac*. All simulations were run sufficiently long to allow a steady state to be achieved, followed by an equally long period with the gradient reversed to check the stability of the spatial pattern. The steady-state concentrations of active PKC at the front and back of the cell are plotted versus *rfrac* (**Fig 2A**). In some simulations, the spatial pattern oscillated, in which case the maximum and minimum values of the oscillation at each end of the cell are plotted. For examples of the simulated time courses, showing the transient behavior with sustained oscillations where applicable, see **S1 Fig and S2 Fig** The PKC activity pattern was reversible for all but one of these simulated conditions. The exception is marked with an asterisk in **Fig 2A**, signifying lack of reversibility here and in subsequent figures of this paper.

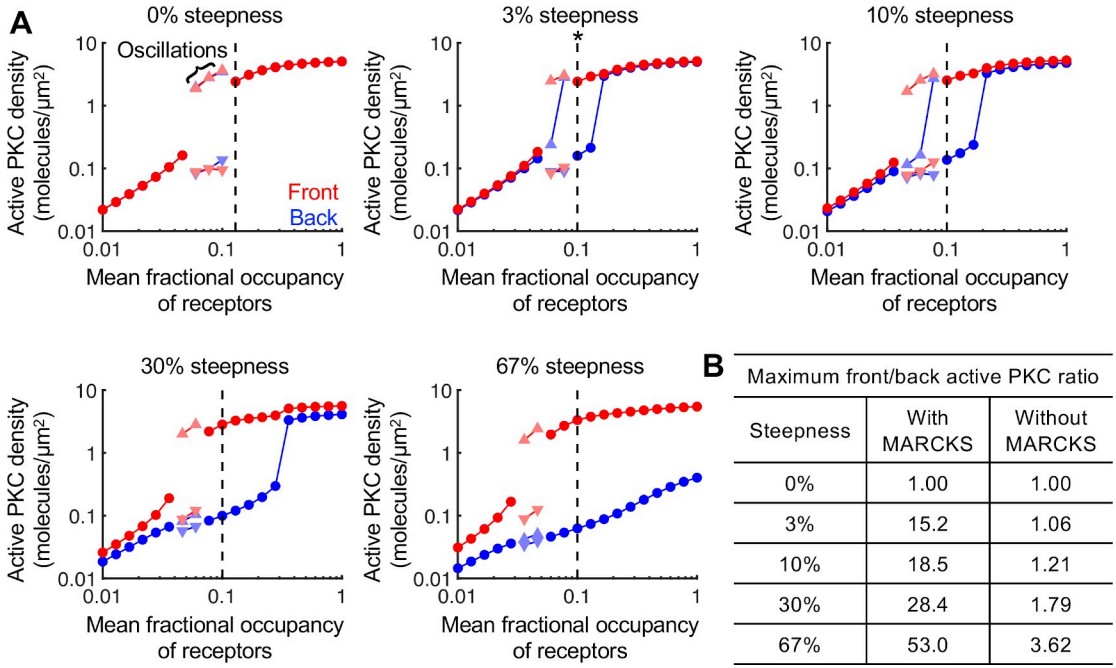

**Fig 2. Gradient amplification by PFL 1 combined with regulation of MARCKS. (A)** The concentration of active PKC molecules at the front (red circle) and back (blue circle) of the cell are plotted as a function of the mean fractional occupancy of receptors, *rfrac*, for varying values of gradient steepness. When the simulations produced oscillations, the maxima (upward-pointing triangles) and minima (downward-pointing triangles) of the oscillations are plotted; the front and back are still denoted by shades of red and blue, respectively. The simulation achieving steady state with the maximum front/back ratio is denoted by the dashed vertical line. For these simulations, the direction of the gradient was reversed after 20,000 s. If the active PKC pattern failed to reverse in response, the simulation is marked with an asterisk. **(B)** Table showing the maximum front/back ratio for each gradient steepness in simulations run with PFL1 and either with or without MARCKS protein.

For simulations that achieved steady state, the degree of polarization is quantified as the ratio of active PKC at the front of the cell over that of the back of the cell, and the value of *rfrac* where this ratio is maximum is marked on each of the plots in **Fig 2A**. For receptor occupancy gradients as shallow as 3%, we found maximum front/back ratios exceeding 10, indicating strong and sensitive polarization of the PLC/PKC pathway. Another quality of the gradient sensing response is its robustness, of which several features might be considered. For example, across the range of % steepness values, the greatest degree of polarization is consistently achieved at approximately the same value of *rfrac* ($\approx 0.1$), and polarized patterns exhibit a consistent density of active PKC at the front of the cell. The aspect of greatest interest to us, however, is the range of *rfrac* values that elicit polarization for a given % steepness, which we refer to as dose-response robustness. As might be expected, dose-response robustness is enhanced as gradient steepness is increased (**Fig 2A**); at a modest steepness of 10% (typical in magnitude of chemotaxis experiments), the range of *rfrac* values for which steady polarization would be evident spans a factor of 2. The range doubles if oscillatory simulations with consistently high front/back ratio are included.

It should be noted that the general inability to polarize the gradient sensing circuit at high *rfrac* is not caused by saturation of receptor occupancy, as would be expected at high concentrations of chemoattractant; in this model, the relative steepness of receptor activation is fixed. Rather, high receptor activation promotes ignition of the positive feedback at the back of the cell as well as at the front.

In previous work [16], it was shown that a single positive feedback was not sufficient for a demonstratively amplified response, i.e., without differential buffering of $PIP_2$ by MARCKS. Accordingly, simulations with PFL 1 but no MARCKS show similar dose responsiveness with respect to *rfrac* but inconsequential polarization (**Fig 2B**). These results identify a promising gradient sensing circuit that combines differential buffering of $PIP_2$ by MARCKS, and the amplification of PLC recruitment by PA, regulation mechanisms supported by evidence in the literature.

## Analysis of the mechanisms driving PLC/PKC polarization

The differential buffering of $PIP_2$ by MARCKS was described in detail previously [16], and it is important to understand this concept in the context of the present model as well. Phosphorylation of MARCKS by active, DAG-bound PKC liberates $PIP_2$ locally, further enhancing DAG production by positive feedback. A key aspect of differential buffering is the maintenance of certain intracellular gradients. In the cytosol, PLC and PKC are close to uniform, whereas there is a gradient of phosphorylated versus unphosphorylated MARCKS in the cytosol. Accordingly, a substantial increase in the diffusivities of cytosolic MARCKS species breaks the polarization of the present model (**Fig 3A**). A large decrease in the diffusivities also decreases polarization, but modestly so. The optimum with respect to diffusivity has been attributed to the ability of dephosphorylated MARCKS to diffuse to the rear of the cell before re-binding $PIP_2$ [16]. The other important gradients are of total and free $PIP_2$ in the plasma membrane. MARCKS-bound $PIP_2$ is protected from hydrolysis, and therefore its spatial range by diffusion is substantial. Thus, net diffusion of MARCKS-bound $PIP_2$ from back to front supplies the front of the cell with substrate, allowing the free $PIP_2$ density to be much higher at the front despite the much higher PLC activity there. Accordingly, reducing the diffusivities of total $PIP_2$ and membrane-associated MARCKS prevents polarization, whereas increasing those diffusivities enhances the extent of polarization (**Fig 3B**).

To further explain the polarization of the present gradient sensing circuit, we developed a steady-state analysis (**S1 Text**), with the simplifying assumption that diffusion of active PLC and DAG-containing species are negligible relative to associated reaction terms. We derived nullcline expressions for the membrane-recruited PLC and for the sum of free and PKC-bound DAG, which we refer to as the *e*- and *d*-nullclines. The intersections of these two curves are fixed points. To construct these curves on a (*d*, *e*) phase plane, certain other variables must be specified. The *e*-nullcline includes the local activated receptor density, *r*, and the cytosolic PLC concentration, *E*. The former is the input to the simulation, with different values at the front and back of the cell, whereas the latter is close to spatially uniform and determined from the simulation. The *d*-nullcline is a straight line with a slope that is inversely proportional to the free $PIP_2$ concentration, with different values at the front and back of the cell determined from the simulation. The phase-plane plot for the case of 10% gradient, *rfrac* = 0.1 from **Fig 2A** illustrates how PFL 1 synergizes with the regulation of MARCKS (**Fig 3C**). PFL 1 is directly responsible for the positive slope of the *e*-nullcline; without PFL 1, the *e*-nullcline has zero slope. In conjunction, regulation of MARCKS is directly responsible for the $PIP_2$ density being higher (lower slope of the *d*-nullcline) at the front versus the back of the cell; without MARCKS, free $PIP_2$ can only be lower, not higher, at the front [16]. This analysis also shows how the situation changes for lower or higher *rfrac*, lending insight into dose-response robustness (**S3 Fig**).

## Subtle asymmetry in cell morphology polarizes the PLC/PKC network with PFL 1 and influences external gradient sensing

Given the demonstrated sensitivity of the present gradient sensing circuit and its dependence on diffusion of MARCKS in the cytosol, we reasoned that a slight asymmetry in cell

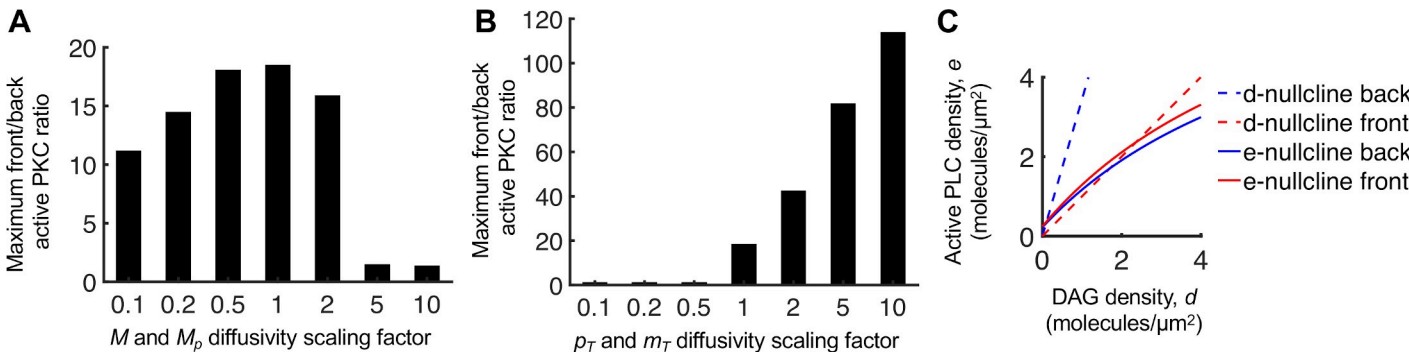

**Fig 3. Analysis of the mechanisms driving PLC/PKC polarization. (A)** Sensitivity of the results in **Fig 2A**, 10% gradient steepness, to the indicated fold-changes in MARCKS ($M$ and $M_p$) cytosolic diffusivities (1 = base case). **(B)** Sensitivity of the results in **Fig 2A**, 10% gradient steepness, to the indicated fold-changes in total PIP$_2$ and membrane-associated MARCKS ($p_T$ and $m_T$) membrane diffusivities (1 = base case). **(C)** Plot of the $d$- and $e$-nullclines (**S1 Text**) evaluated at the front and back of the cell for the base-case parameters as in **Fig 2A**, with 10% gradient steepness and $rfrac$ = 0.1.

morphology might be a sufficiently strong spatial cue to polarize signaling in response to uniform receptor occupancy. To test this, the cell geometry was altered such that the back end of the cell was blunted compared to the front end (**Fig 4A**). With uniform stimulation and the same parameter set used in **Fig 2A**, the shape asymmetry polarized the signaling network, disfavoring the blunted end, for a range of $rfrac$ values (**Fig 4B**). As expected, this polarization requires a gradient of phosphorylated MARCKS, with a dependence on cytosolic MARCKS diffusivities similar to polarization induced by an external gradient (**Fig 4C**, compare to **Fig 3A**).

The tendency of the asymmetric cell geometry to polarize was also tested with receptor occupancy gradients in either direction (**Fig 4D**). Consistent with the rest of the paper, the front of the cell refers here to the right end, with the highest receptor occupancy, and the back of the cell is at the left end, with the lowest receptor occupancy. With the blunt end of the cell at the back, the maximum degree of active PKC polarization is only modestly affected by receptor occupancy gradients of 3% and 10%, relative to uniform stimulation, whereas 30% and 67% gradients elicit substantially greater polarization similar to the symmetric geometry (compare **Fig 4D** and **Fig 2B**). Consistent with that trend, when the blunt end of the cell was at the front (right-to-left gradient), the shape effect opposes the influence of the receptor occupancy gradient and dominates for 3% and 10% gradients, whereas the steeper gradients were sufficient to overcome the geometry effect. These results indicate that the cell's local morphology can readily alter or play a dominant role in the polarization of PLC/PKC signaling.

## The model predicts a critical role of DAG kinases affecting the responsiveness of the gradient sensing network through PFL 1

Having characterized and explained how PFL 1 influences polarization of the signaling network, we considered the robustness of this system to substantial changes in rate parameters. Each of the rate constants and affinity parameters was increased by 3X and reduced to 0.3X to yield a full order-of-magnitude range, and the analysis shown in **Fig 2A** was repeated for each. A plot of the maximum front/back ratios of activated PKC shows that polarization is fairly robust to a 3-fold change of each parameter in at least one direction (**S4 Fig**). The changes that break the polarization are those that weaken PLC recruitment, alter MARCKS phosphorylation by PKC (as explored in [16]), or alter lipid metabolism. The three parameters associated with the latter ($k_{DAGK}$, $k_{PAP}$, and $k_{basal,dp}$) are pseudo-first-order rate constants that reflect enzymatic metabolism of DAG and PA. Reasoning that the interconversion of DAG and PA is

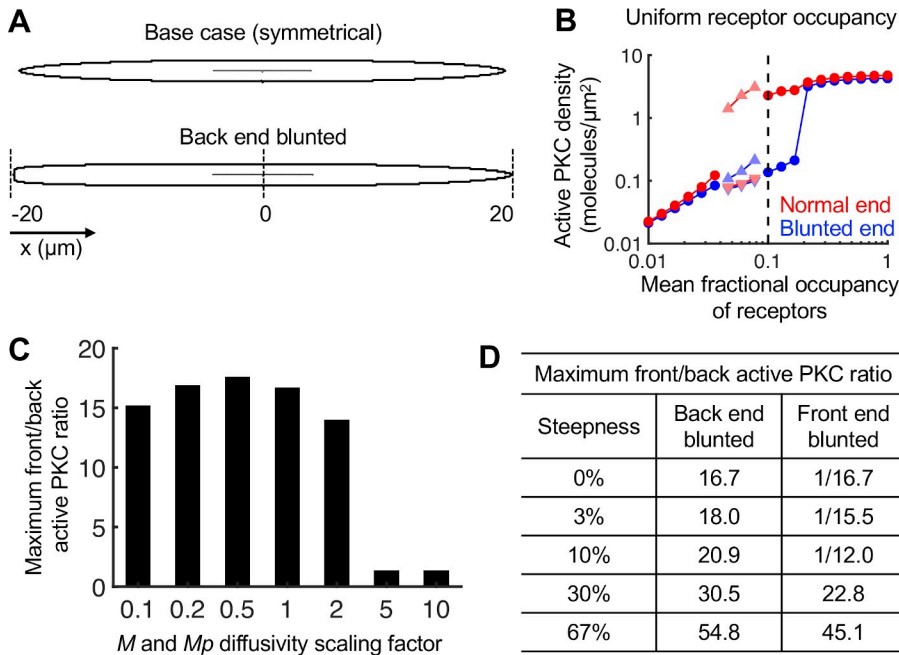

**Fig 4. Subtle asymmetry in cell morphology causes spontaneous polarization of the proposed PLC/PKC network and influences external gradient sensing. (A)** Comparison of the symmetric cell geometry versus the asymmetric cell geometry. **(B)** Results for the asymmetric geometry with uniform receptor occupancy (0% gradient steepness) show spontaneous polarization of the system for a particular range of receptor occupancy values. Symbols have the same meanings as in **Fig 2A**. **(C)** Sensitivity of the results in **B** to the indicated fold-changes in MARCKS ($M$ and $M_p$) cytosolic diffusivities (1 = base case). **(D)** Table showing the maximum front/back active PKC ratio for each gradient steepness indicated using the asymmetric geometry. Consistent with the rest of the paper, the front and back of the cell refer to the ends with the highest and lowest receptor activation, respectively. For the front-end blunted simulations where the ratio is less than 1 (internal gradient opposite the external gradient), the inverse of the ratio is indicated.

a pivotal aspect that needs to be understood, we devised a simple steady-state analysis assuming negligible diffusion of DAG and PA (**S1 Text**). This analysis predicts a consistent proportional relationship between the concentrations of DAG and PA. To verify this conclusion, the steady-state concentration of PA at the front of the cell was plotted versus that of DAG for all of the simulations analyzed in **Fig 2A**; each of these points lies approximately on the line predicted by the analytical expression (**Fig 5A**).

The analysis shows that the activity of DAG kinases, reflected in the value of the rate constant, $k_{DAGK}$, directly alters the PA/DAG ratio. Intuitively, diminishing DAG kinase activity is expected to result in higher DAG and active PKC levels; however, in the model with PFL 1 and a nominal 10% gradient of receptor occupancy, a reduction of $k_{DAGK}$ to 0.3 times its base value ablates polarization, whereas an increase of $k_{DAGK}$ to 3 times its base value substantially enhances the degree of polarization and dose-response robustness (**Fig 5B**). Analysis of DAG and PA levels for 1x and 3x $k_{DAGK}$ (**Fig 5C**) shows that as the parameter is increased, the density of DAG at the cell front is decreased as expected, but only modestly so (~30% reduction) because of the substantially increased abundance of PA; at the rear of the cell, the impact on DAG is relatively far greater (~70% reduction). These effects of increased $k_{DAGK}$ extend to receptor-activation gradients with 3% steepness (**Fig 5D**).

To further test this analysis, we predicted that we could increase the steady-state levels of both DAG and PA, while maintaining the same PA/DAG ratio, by proportional reduction of all three of $k_{DAGK}$, $k_{PAP}$, and $k_{basal,dp}$. To normalize for these changes at the level of PFL 1 and PKC activity, we applied the same reduction factor to the PA-PLC affinity parameter, $K_{PA}$, and

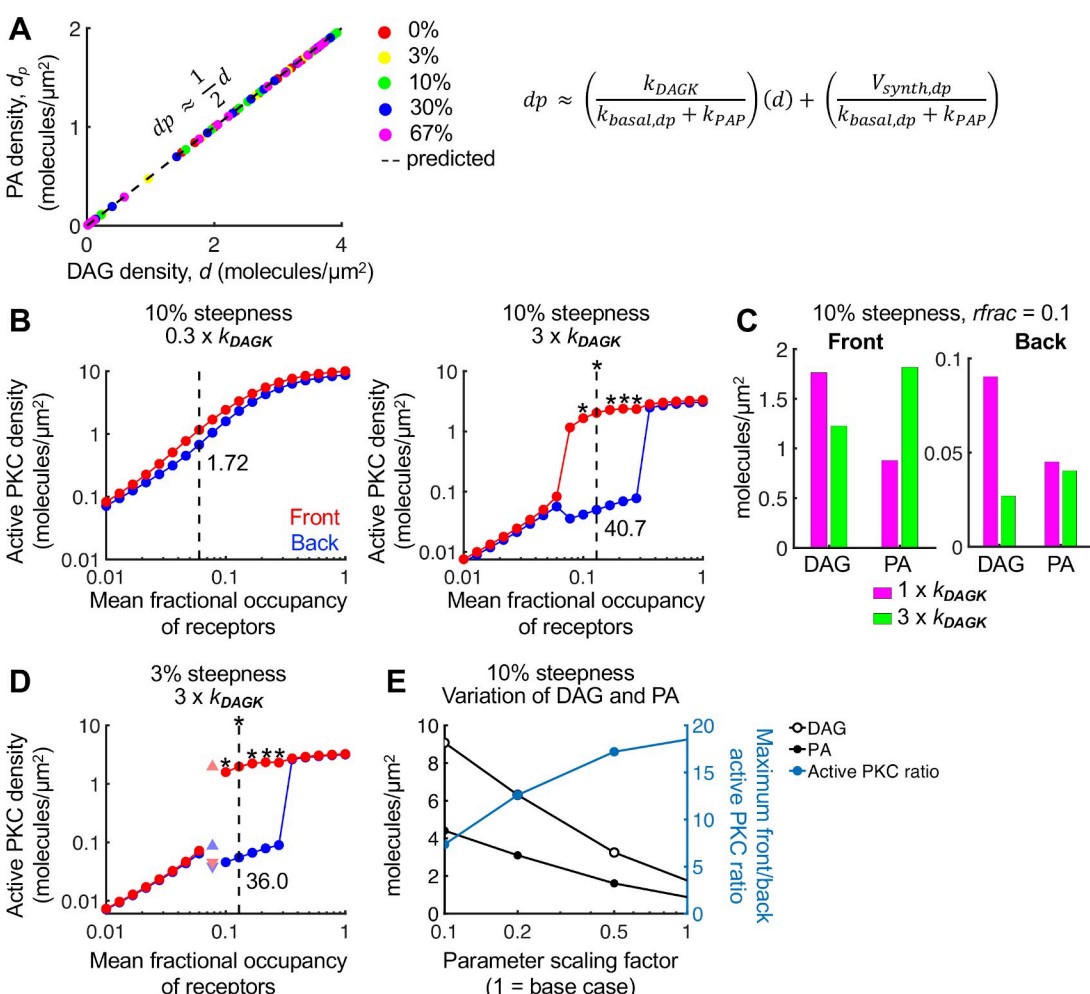

**Fig 5. Variation of PA/DAG ratio modulates the sensitivity and dose response of the pathway. (A)** The final steady-state concentration of PA ($d_p$) is plotted versus the concentration of DAG ($d$) at the front of the cell for each simulation presented in **Fig 2**. These points fall along the line predicted by the equation shown, for which net diffusion of lipids is assumed to be slow. **(B)** Relative to the parameter set associated with **Fig 2**, the parameter $k_{DAGK}$ was taken at 0.3 or 3 times its base value, at 10% gradient steepness. Colors and symbols have the same meanings as in **Fig 2A**. The maximum front/back ratio is shown underneath the symbols. **(C)** Bar plots comparing the DAG and PA steady-state concentrations at the front and back of the cell for 1x and 3x $k_{DAGK}$ simulations, with 10% steepness and *rfrac* = 0.1. **(D)** Same as **B**, but with 3% gradient steepness. For **B-D**, the direction of the gradient was reversed after 20,000 s. If the active PKC pattern failed to reverse in response, the simulation is marked with an asterisk. **(E)** The parameter scaling factor refers to the fold-change by which the parameters $k_{DAGK}$, $k_{PAP}$, $k_{basal,dp}$, and $K_{PA}$ were decreased, while the parameter $k_{off,c}$ was divided by the scaling factor to increase its value. These parameter changes increase DAG and PA levels at the front of the cell systematically, while maintaining approximately the same PA/DAG ratio and comparable effects of PA and DAG on PFL 1 and PKC recruitment, respectively.

to the DAG-PKC affinity (by increasing $k_{off,c}$). With these changes, we were able to maintain polarization while increasing DAG and PA levels by almost an order of magnitude (**Fig 5E**).

## PKC-mediated activation of PLD (PFL 2) confers less responsiveness than PFL 1, but the two feedbacks can synergize in the polarization of PLC/PKC signaling

The other putative source of feedback we consider is the ability of active PKC to amplify the hydrolysis of phosphatidylcholine by PLD, producing PA and additional DAG upon

dephosphorylation of PA (PFL 2). Parameters characterizing PFL 2 are a saturation constant, $K_{PLD}$, gain parameter, $\gamma$, and Hill coefficient, $n$. Extensive parameter sweeps of $K_{PLD}$ and $\gamma$ were performed with $n$ fixed at either 1 (Michaelean sensitivity) or 2 (modestly ultrasensitive). At a 10% gradient in receptor activation, PFL 2 along with the regulation of MARCKS was unable to polarize DAG and active PKC with $n = 1$, whereas with $n = 2$, a particular combination of $K_{PLD}$ and $\gamma$ values yielded strong polarization; however, the degree of polarization and dose-response robustness are less than those with PFL 1 instead (**Fig 6A**). That said, ultrasensitive PFL 2 shows synergy when combined with PFL 1. Firstly, the inclusion of both PFLs (with re-optimization of $K_{PLD}$ and $\gamma$ for PFL 2) allowed for polarization without MARCKS (**Fig 6B**). When all three feedbacks (PFLs 1 and 2 and regulation of MARCKS) were present, both the degree of polarization and the dose-response robustness were dramatically enhanced, at the expense of reversibility (**Fig 6C**). Consistent with these conclusions, polarization of the system with PFL 2 is broken by 3-fold changes in parameters that influence PA levels ($k_{DAGK}$ or $K_{PLD}$) except when all three feedbacks are included (**S5 Fig**).

### Thresholds for chemotactic migration matching those of DAG/PKC polarization yield efficient collective invasion and chemotactic wavelets in simulations of wound invasion

Having characterized the polarization of the proposed PLC/PKC network, we asked how it might influence directed cell migration in a physiological context. To address this, we adapted a hybrid simulation of wound invasion, in which the concentration of PDGF evolves as a continuum in one dimension, and fibroblasts are treated as motile line segments [28]. A key prediction of the previous wound invasion model, which did not consider a polarizable gradient sensing mechanism, is that the cells can collectively generate a PDGF gradient through receptor-mediated endocytosis and lysosomal degradation [29], a concept that has since been verified in experimental systems [30–33]. Thus, fibroblast invasion of the simulated wound is guided by a chemotactic wave. This model was modified such that chemotaxis is switched on or off according to the receptor activation conditions that yield polarization of PKC activity for the modestly robust model and base-case parameters used to generate the results shown in **Fig 2**. When chemotaxis is switched off, the cell moves in a random direction (left or right). The cell migration parameters were chosen to be consistent with experimentally measured migration speed and chemotactic (forward migration) index values [10,11].

The movements of the cells in the simulation may be animated to visualize the collective behavior (**Fig 7A** and **S1 Movie**), and the progress of the cell population shows substantially faster invasion relative to simulations in which chemotaxis was not allowed (random migration only) (**Fig 7B**). This was anticipated because of the chemotactic behavior, but inspection of the simulations revealed an emergent property of the system. While the major chemotactic wave at the leading front of the cell population, located within the steepest gradient in PDGF concentration, remains a primary feature (**Fig 7C**), the present model predicts the existence of chemotactic wavelets that arise and propagate within the plateau region of the PDGF concentration profile (**Fig 7D**). Thus, in the present model, significant chemotaxis occurs throughout the wound (**Fig 7C & 7D**), as cells that enter the simulated wound at later times are able to create and follow the mesoscopic waves to explore more of the space (**S1 Movie**). These results show that even modest dose-response robustness of the circuit is sufficient to affect chemotaxis in tissues, where chemoattractant gradients are dynamic. Indeed, it is the dynamic nature of the gradients that makes this possible.

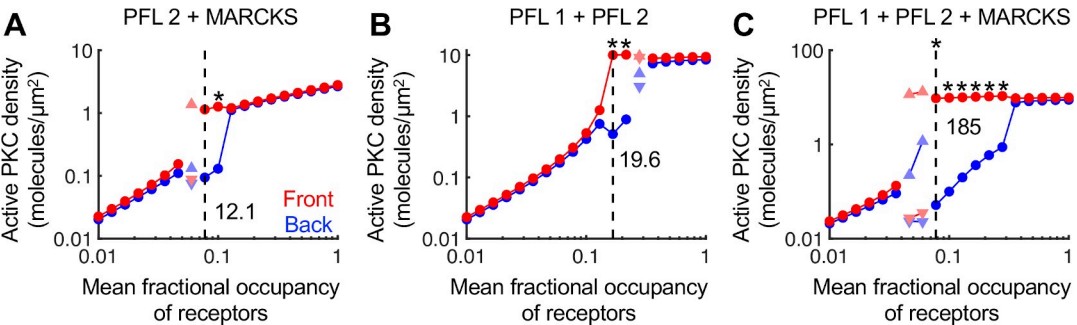

**Fig 6. PFL 2 confers less responsiveness than PFL 1, but the two can synergize in the polarization of PLC/PKC signaling.** The simulations shown are all at a 10% gradient of receptor occupancy. All colors, symbols, and values have the same meanings as in previous figures. **(A)** PFL 2 and MARCKS regulation but no PFL 1. For PFL 2, the parameters $K_{PLD}$ and $\gamma V_{synth,dp}$ are set at 1 and 1, respectively. **(B)** PFL 1 and PFL 2 but no MARCKS. The parameters $K_{PLD}$ and $\gamma V_{synth,dp}$ are set at 0.1 and 10, respectively. **(C)** All three feedbacks are included, and $K_{PLD}$ and $\gamma V_{synth,dp}$ are set as in **B**. For all of these simulations, the direction of the gradient was reversed after 20,000 s. If the active PKC pattern failed to reverse in response, the simulation is marked with an asterisk.

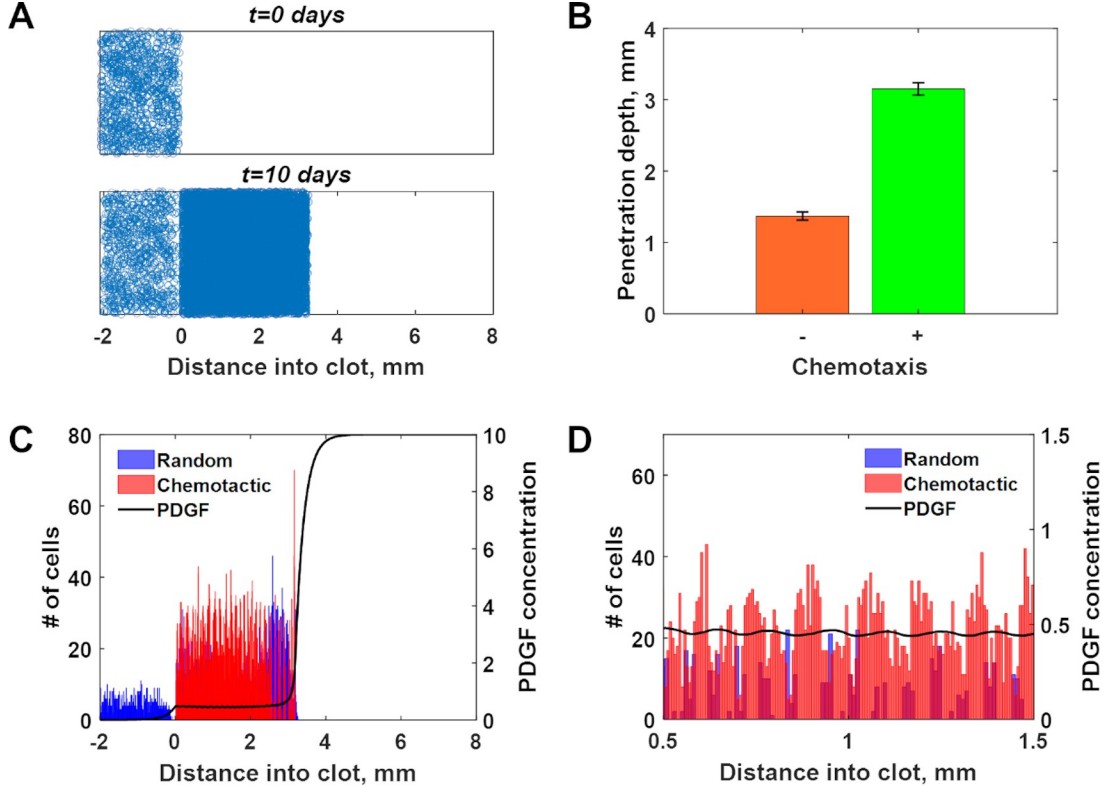

**Fig 7. Simulation of wound invasion with chemotactic switching based on DAG/PKC polarization. (A)** Depiction of individual cells in the hybrid simulation, in which the concentration of PDGF is modeled as a continuum. The cells are initially seeded in the adjacent dermis; position along the vertical dimension is for visualization only. As time elapses, the cells invade and populate the clot region through a combination of directed migration and proliferation (see also **S1 Movie**). **(B)** Depth of penetration into the wound at $t = 10$ days for random migration only or with chemotaxis allowed (mean ± s.d., $n = 10$ simulations each). **(C)** Spatial profiles of cell densities (randomly or chemotactically migrating) and dimensionless PDGF concentration. **(D)** Zoomed-in view of the cell density and PDGF concentration profiles for 1 mm length of the clot region, showing chemotactic wavelets.

## Discussion

### Novel ways that lipid signaling might enhance chemotactic gradient sensing, corroborated by modeling

While the bulk of the signal transduction literature focuses on proteins and their regulation by phosphorylation, it is easy to forget the significance of lipid second messengers. In the context of spatial sensing and directed cell migration, phosphoinositides and other plasma membrane lipids are uniquely positioned, owing to their peripheral location, slow diffusion, relative abundance, and diversity of lipid-protein interactions [14,34]. For more than two decades, research on lipid-mediated mechanisms of chemotactic gradient sensing has emphasized signaling through class I phosphoinositide 3-kinases (PI3Ks), which phosphorylate $PIP_2$ to produce phosphatidylinositol (3,4,5)-trisphosphate [35,36]. That lipid and its breakdown product, phosphatidylinositol (3,4)-bisphosphate, interact with a host of signaling proteins that locally enhance F-actin polymerization by affecting activation of the Arp2/3 complex [37]. However, in most cell types tested, PI3K signaling is not absolutely required for chemotaxis, and, in fibroblasts and macrophages, even the Arp2/3 complex is dispensable for chemotaxis [10,11]. For chemotaxis of fibroblasts and other mesenchymal cells, PLC/PKC signaling is required, and so increased focus on this pathway and on DAG as a lipid second messenger is warranted.

The present models consider the dynamics of lipid signaling beyond DAG. Plasma-membrane DAG is phosphorylated by DAG kinases to produce PA, a lipid second messenger in its own right [17,18]. With PFL 1, we considered the ability of PA to enhance the observed affinity of PLCγ for $PIP_2$ [23], presumably by increasing the lifetime of enzyme binding to membranes. We note that PLCβ1 activity, which is fostered by G protein-coupled receptor signaling, is also promoted by PA [38]. In the model, this putative feedback locally enhances PLC recruitment by activated receptors and the hydrolysis of $PIP_2$. This effect of PA at the cell's leading edge offsets the action of DAG kinases in metabolizing DAG. As a single feedback, PFL 1 was not sufficient to polarize signaling in the model, requiring also the recruitment and activation of PKC by DAG. The necessary action of PKC could involve either or both of two mechanisms. The more potent of the two is the neutralizing phosphorylation of MARCKS by PKC, which directly synergizes with the enhancement of PLC recruitment by enhancing the supply of $PIP_2$ substrate. The less direct route is PFL 2, by which PKC enhances the activity of PLD; PLD hydrolyzes phosphatidylcholine to produce PA and thus presents a parallel pathway for DAG generation. Highlighting the weaker influence of PFL 2 on polarization in the model, it was necessary to add positive cooperativity with respect to active PKC and to fine-tune the values of the PFL 2-associated parameters. As one might expect, including all three feedback mechanisms in the model, with the provisions for PFL 2, yielded the most sensitive and robust gradient sensing circuit.

### Robustness and reversibility of polarization

The key challenges met to varying degrees by our models is to achieve polarization in shallow gradients and for an appreciable range of chemoattractant concentration (dose-response robustness). These two aspects go hand in hand for our models: greater sensitivity and degree of polarization were generally accompanied by greater dose-response robustness. These positive aspects could be enhanced, through changes to the parameter values or addition of a feedback, but with loss of reversibility; the system is polarized and tends to remain locked in place even when the gradient is reversed. This observation is consistent with analyses of other gradient sensing models that show a tradeoff between amplification and self-locking behavior [39–42]. Self-locking is generally considered undesirable, because it precludes a prompt response

to dynamic changes in the chemoattractant gradient. To adjust the directionality of movement, a cell with its chemotactic signaling polarity locked would have to execute a turn. For a gradual response such as wound invasion, such a system might be adequate or even advantageous. Obviously, with a 1D wound geometry, persistent chemotaxis would yield the maximum rate of invasion. But even with instantaneous switching between polarization states, our hybrid invasion model showed efficient invasion, with chemotactic waves arising throughout the volume of the wound occupied by cells.

As suggested previously, a gradient sensing circuit capable of polarizing in response to shallow gradients can also polarize in response to an asymmetric cell geometry [16]. This is not a generality, however. In the context of this model, it is particular to the proposed regulation of MARCKS, where the local ratio of plasma membrane area to cytosolic volume affects the dynamics. This follows basic ideas discussed in detail elsewhere [43–47]. If sensitivity to morphology were significant, it could act as an additional feedback mechanism. On the other hand, from the perspective of sensing external gradients, any such intrinsic cue might be considered a detriment.

## Testable hypotheses guided by the models

Analysis of the present models offers a guide for experiments designed to test certain predictions, with polarization of DAG in chemotaxing cells as the essential readout. At a phenomenological level, a basic prediction of the models is that there is a minimum threshold of receptor activation, determined by the chemoattractant concentration, below which the pathway cannot polarize. Considering the heterogeneity in a cell population, for a suitable gradient one should expect to find subpopulations that do or do not show DAG polarization. For the shallow gradients typically achieved in chemotaxis chambers, an upper bound on receptor activation is also predicted. Whether or not either the minimum threshold or upper bound can be exceeded would depend on a cell's expression level of the cognate receptor.

Regarding molecular mechanisms, the most basic prediction is the requirement for PKC activity, which may be readily tested using isoform-selective inhibitors. The regulation of MARCKS requires PKC activity, whereas modulation of PLD requires interaction with membrane-associated PKC but not PKC kinase activity [48,49]. From there, perturbations affecting MARCKS and PLD may be pursued. The line of experimentation outlined above does not address PFL1, which does not rely on PKC. The prediction most directly related to PFL 1 concerns the role of DAG kinases, which consume DAG but produce PA in the process. It is predicted that even a partial inhibition of DAG kinase activity (achieved either pharmacologically [50,51] or by depleting DAG kinase isoforms), while increasing DAG on a whole-cell level, would abrogate polarization for all doses of chemoattractant. Conversely, enhancement of DAG kinase activity, which might be achieved by overexpressing one or more of the isoforms, is predicted to enhance the dose-response robustness of polarization by increasing the upper bound of receptor activation. If true, more cells would be able to polarize at higher concentrations of chemoattractant.

Finally, the effect of cell geometry could be tested by co-expression of or labeling with a fluorescent, cytosolic volume marker. Imaged by epifluorescence, the marker would quantify the local height of the cytoplasm; the lower this quantity in a lamellipod, the higher the membrane area/volume, and the more likely the DAG is to be enriched there if the predicted geometry effect is significant.

## Limitations of the models

The purpose of these models is to provisionally test the plausibility of the proposed feedback mechanisms. Modeling of stochastic effects and three-dimensional cell geometries with

moving boundaries would make the simulations more realistic, with far greater computational expense. The topology of the signaling network is also by no means complete. As already noted, certain details of the mechanisms considered are incompletely understood, and other mechanisms not considered here might also contribute. Regarding the latter, one might consider other possible roles of the lipids involved. Whereas we focused on the putative influence of PA on PLC, PA has also been reported to enhance phosphatidylinositol 4-phosphate 5-kinase activity [52], which could boost the rate of $PIP_2$ generation. In principle, we would expect this effect to amplify PFL 1. $PIP_2$ also functions in other capacities that could impact PLC/PKC signaling and chemotaxis. For one, $PIP_2$ is the preferred substrate of class I PI3Ks. Although PI3K signaling is not absolutely required for mesenchymal chemotaxisFormatting. . . please wait, PI3K competes with PLC for the common substrate and could be influenced by MARCKS [53]. $PIP_2$ is also bound by the actin-modifying proteins cofilin and profilin at the plasma membrane [54]. Local modulation of free $PIP_2$ and its hydrolysis by active PLC is expected to impact cofilin and profilin functions in chemotaxis [55,56]. Finally, we note that $PIP_2$ is a cofactor for PLD, influencing the enzyme's membrane localization and activity [57].

One of the key experimental observations that we have yet to adequately explain is the ability of phorbol ester (a DAG mimic) to elicit fibroblast chemotaxis when presented as a shallow gradient, even in PLCγ1-null cells [11]. This scenario is akin to the model with PFL 2 only and no MARCKS, for which we could not find any set of parameter values that yielded polarization of PKC activity in response to a shallow ($\leq$ 10%) gradient. Whether or not a phorbol ester gradient polarizes DAG/PKC is presently unknown; it is possible that chemotaxis occurs regardless. If polarization does occur in PLC-null cells executing phorbol ester chemotaxis, the nature of the underlying mechanisms will need to be clarified.

## Materials and methods

### Models of PLC/PKC signaling

The reaction-diffusion models are composed of partial differential equations and associated boundary and initial conditions. The species in the models and their interactions are described here; mathematical details and justifications for chosen parameter values are given in **S1 Text**. Within that document, model species, their diffusivity values, and initial conditions are listed in S1 Table, and rate equations and base-case values of rate constants are listed in S2 Table. Cytosolic species, signified by capital letters, have local concentrations in μM, whereas membrane species, signified by lowercase letters, have local densities in #/μm$^2$. Rather than explicitly model the ligand-receptor dynamics, in this work we assume a linear profile of active receptors ($r$) as the model input, calculated from **Eq 1**.

$$r = 130\ rfrac\left[1 + rsteep\left(\frac{x}{40}\right)\right] \qquad (1)$$

As explained under Results, the prefactor determines the midpoint value of $r$; when the dimensionless $rfrac$ = 1, the midpoint value of $r$ = 130/μm$^2$, corresponding to approximately 1 x 10$^5$/cell. The distance from the midpoint, $x$, is in μm, and it is scaled by the cell length of 40 μm. Therefore, $rsteep$ is the fractional gradient steepness; a value of $rsteep$ = 0.1, for example, corresponds to a 10% difference between the values of $r$ at the extreme front and back of the cell.

Referring to the diagram shown in **Fig 1A**, activated receptors recruit inactive PLC enzyme ($E$) from the cytosol to form active PLC at the membrane ($e$). Active PLC hydrolyzes $PIP_2$ (free-density $p$). In the absence of stimulation, the density of $PIP_2$ is maintained through basal synthesis and turnover. $PIP_2$ hydrolysis by PLC generates the lipid second messenger, DAG

($d$). DAG is phosphorylated by DAG kinases to produce PA ($d_p$), which can be dephosphorylated by phosphatidic acid phosphatases to recover DAG [22,58]; these are modeled as pseudo-first-order reactions. Like PIP$_2$, PA is also subject to basal generation and consumption; however, basal generation of PA is intentionally very low. DAG recruits catalytically competent but inactive PKC ($C$) from the cytosol by reversible binding of the tandem C1 domain of PKCs, forming active PKC ($c^*$) at the membrane [59]. Membrane-bound PKC is sensitive to dephosphorylation, and so the active PKC is converted to an inactive, membrane-bound form ($c$); this species is either autophosphorylated to regenerate active PKC or it dissociates to join the cytosolic pool. MARCKS is an abundant substrate of PKC that is present in both cytosolic and membrane-bound forms [14]. The unphosphorylated, cytosolic form ($M$) inserts into the plasma membrane via its myristoyl group and interacts with PIP$_2$ via its effector domain; the latter is a high-avidity, electrostatic interaction, and thus, MARCKS sequesters a substantial fraction of the intracellular PIP$_2$ [13,60]. The membrane-bound forms of MARCKS ($m$) are phosphorylated by active PKC, causing loss of affinity for the plasma membrane and liberation of PIP$_2$; the phosphorylated form of MARCKS ($M_p$) is cytosolic [12]. MARCKS is dephosphorylated in the cytosol by a pseudo-first-order reaction to complete the cycle.

Imposed on this basic network structure are two additional PFLs. As described under Results, PFL 1 considers that the PA engages PLC in complex with activated receptors and thus extends the enzyme's mean lifetime (or, reduces its effective off-rate) at the plasma membrane. The net rate of PLC recruitment, $V_{PLC}$, is given by **Eq 2**.

$$V_{PLC} = k_{on,e}(r - e)E|_S - k_{off,e}\left(\frac{1 + \varepsilon K_{PA}d_p}{1 + K_{PA}d_p}\right)e \tag{2}$$

In this equation, the parameters characterizing PFL 1 are $K_{PA}$, the equilibrium constant of PA-PLC interaction, and $\varepsilon$, a dimensionless escape probability (see **S1 Text**). With either $K_{PA}$ set to zero or $\varepsilon$ set to 1, PFL 1 is turned off. PFL 2 considers that active PKC engages and enhances the activity of PLD. This influence on the rate of PA synthesis, $V_{PLD}$, is modeled as a Hill function in **Eq 3**.

$$V_{PLD} = V_{synth,dp}\left(\frac{1 + \gamma(K_{PLD}c^*)^n}{1 + (K_{PLD}c^*)^n}\right) \tag{3}$$

In this equation, the parameters characterizing PFL 2 are $K_{PLD}$, a saturation constant, $\gamma$, a dimensionless gain parameter, and $n$, the Hill coefficient. With either $K_{PLD}$ set to zero or $\gamma$ set to 1, PFL 2 is turned off.

## Implementation of PLC/PKC models

The partial differential equation models are implemented in Virtual Cell (http://www.vcell.org), a computational environment for modeling and simulation in cell biology [61]. The Biomodel and primary simulations are publicly available in Virtual Cell under user name jnosbis, Biomodel name 'Nosbisch chemotaxis 2020'. As in Mohan et al. [16], the base two-dimensional geometry is an ellipse with a major axis length of 40 μm and minor axis length of 1.8 μm (**Fig 1B**); these dimensions were selected according to the important dimensions of a migrating fibroblast. For the asymmetric geometry considered in **Fig 4**, the left end of the ellipse was blunted by changing the $y$-component of the ellipse equation to have an exponent of 6, rather than 2, for negative values of $x$.

Virtual Cell uses a finite-volume method to numerically solve the reaction-diffusion equations, and the Fully-Implicit Finite Volume (variable time step) solver was used, with voxel

dimensions of $\Delta x = \Delta y = 0.1$ µm and a maximum time step of 0.1 s. We confirmed that the computational results were not significantly affected by modest changes to those values. The initial conditions are such that the system is stationary in the absence of stimulation. Simulations were run for 20,000 s to ensure either a steady state or sustained oscillations. For all simulations, the reversibility of the spatial pattern was assessed by reversing the gradient of PDGF receptor activation across the cell after a steady state was reached and extending the simulation for another 20,000 s. If the spatial pattern achieved a steady state but failed to reverse, then we considered that simulation to display locking behavior, and the simulation is marked with an asterisk.

## Wound healing model

The model of collective cell migration is adapted from the hybrid simulation strategy of Monine and Haugh, described in detail previously [28]. Briefly, the dimensionless PDGF concentration, $u$, is treated as a continuum, subject to synthesis (only in the clot), interstitial diffusion, intrinsic degradation, and degradation by cells (receptor-mediated endocytosis). Local receptor activation, $r$, is given by the algebraic function, $r = u^2/(1 + u + u^2)$, based on a quasi-steady-state approximation [29]; therefore, $u = 1$ corresponds to $r = 1/3$. The local value of $r$, along with the calculated cell density, determines the cell-mediated degradation of PDGF and the net rate of cell proliferation or death. The original code was adapted from C to MATLAB; the version that was implemented to generate the results shown in **Fig 7** is provided in the **Supporting Information (S2 Text)**. The difference in the new model is that the migration behavior, rather than depending on intracellular signaling in an analog fashion, switches between random and chemotactically biased states. In the random state, the cells move with a diffusivity $D_{v1} = 3\text{x}10^{-4}$ mm$^2$/h. Considering a persistence time of $\sim 0.5$–$1$ h for fibroblasts, this corresponds to a cell speed of $\sim 0.03$ mm/h ($\sim 0.5$ µm/min) [62]. In the chemotactic state, the cell movement is governed by diffusion and convection. For the base case, the chemotactic (convective) velocity in the direction of the PDGF gradient, $S_{tax}$, was chosen as 0.03 mm/h to match the cell speed estimate given above. The random component of the chemotactic cells' movements was set to a lower value of $D_{v2} = 1\text{x}10^{-4}$ mm$^2$/h to represent noise in the cell's movement up-gradient. It is envisioned that the cells' movements in the other two orthogonal dimensions of the tissue would be comparable regardless of migration state, and therefore a rough estimate of the associated chemotactic (forward migration) index is $\sim 1/3$, a modest value considering that measured population estimates of this forward migration index are $\approx 0.2$ and include tracks of chemotaxing and non-chemotaxing cells [11].

The switching between random and chemotactic migration were taken from the approximate cut-offs between non-polarizing and polarizing conditions indicated in **Fig 2**, supplemented with additional data for $rsteep = 0.2, 0.4$, and $0.5$. For each value of $rsteep$, the lower and upper bounds of $rfrac$ were approximated (at $rsteep = 0.67$, it was found that the upper bound, if it exists, is $>> 1$), with oscillations considered non-polarizing. These data were used to construct a relationship between the value of $rfrac$ (the variable $r$ calculated in the wound healing model) and the corresponding minimum value of $rsteep$ required for polarization; in the wound healing model, $rsteep$ was equated to $\Delta r/r$, where $\Delta r$ is the difference in $r$ values evaluated +/- 20 µm from the midpoint (matching the cell length in the PLC/PKC models). The relationship was fit to **Eq 4**, reminiscent of an inter-molecular potential.

$$\left(\frac{\Delta r}{r}\right)_{min} = 0.01 + a\left[\left(\frac{r_{opt}}{r}\right)^n - \frac{n}{m}\left(\frac{r_{opt}}{r}\right)^m + \frac{n}{m} - 1\right] \tag{4}$$

The first term, 0.01, is an homage to the notion that the limit of spatial gradient sensing is a

~ 1% difference across a cell's length [63]. The other constant parameter values are $a = 1.7$, $r_{opt} = 0.12$, $n = 2$, and $m = 1.5$. A plot of Eq 4 with these parameter values is shown in S6 Fig

## Supporting information

**S1 Text. Modeling details.**
(PDF)

**S2 Text. MATLAB code used to simulate wound invasion (Nosbisch_et_al_wound_sim.m).**
(M)

**S1 Fig. Time courses of all model species for Fig 2A, 10% steepness and *rfrac* = 0.1.**
(PDF)

**S2 Fig. Transition from oscillations to stable pattern as *rfrac* is increased (10% steepness).**
(PDF)

**S3 Fig. Phase plots for different *rfrac* values.** These correspond to the simulations analyzed in Fig 2A with 10% gradient steepness. The plot with *rfrac* = 0.1 is the same as in Fig 3A. When *rfrac* = 0.0278, DAG is low at both the front and back of the cell; with *rfrac* = 0.278, DAG is high at both the front and back of the cell.
(PDF)

**S4 Fig. Systematic variation of kinetic parameters (no PFL 2).**
(PDF)

**S5 Fig. Variation of key parameters with PFL 2 included in the system.** The parameters targeted were $k_{DAGK}$, which affects the conversion of DAG to PA; $K_{PA}$, which affects the PA-PLC affinity in PFL 1; and $K_{PLD}$, which affects the sensitivity of PFL 2 to active PKC. Each parameter was decreased to 0.3X and increased to 3X. Each plot shows active PKC density vs. *rfrac* for 10% gradient steepness, as in Fig 6.
(PDF)

**S6 Fig. Plot of Eq 4.** For each cell $i$ in the simulation, its mean receptor activation $r_i$ and the difference in receptor activation across its length $\Delta r_i$ were calculated. Cells with receptor activation states above the curve engaged in chemotaxis, whereas those with states below the curve engaged in random migration only.
(PDF)

**S1 Movie. Animation of the simulation depicted in Fig 7A.**
(AVI)

## Acknowledgments

We acknowledge use of the freely available Virtual Cell software (supported by NIH Grant Number R24 GM134211 from the National Institute for General Medical Sciences).

## Author Contributions

**Conceptualization:** Krithika Mohan, Timothy C. Elston, James E. Bear, Jason M. Haugh.

**Data curation:** Jamie L. Nosbisch, Anisur Rahman.

**Formal analysis:** Jamie L. Nosbisch, Anisur Rahman, Jason M. Haugh.

**Funding acquisition:** Timothy C. Elston, James E. Bear, Jason M. Haugh.

**Investigation:** Jamie L. Nosbisch.

**Writing – original draft:** Jamie L. Nosbisch, Jason M. Haugh.

**Writing – review & editing:** Jamie L. Nosbisch, Anisur Rahman, Krithika Mohan, Timothy C. Elston, James E. Bear, Jason M. Haugh.

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
