## [Decision Letter · Decision Letter 0]

13 Sep 2019

Dear Dr Haugh,

Thank you very much for submitting your manuscript 'Mechanistic models of PLC/PKC signaling implicate phosphatidic acid as a key amplifier of chemotactic gradient sensing' for review by PLOS Computational Biology. Your manuscript has been fully evaluated by the PLOS Computational Biology editorial team and in this case also by independent peer reviewers.

The reviewers appreciated the attention to an important problem and appreciate the soundness of the model and the insights gained from it. However, they raised substantial concerns about the manuscript as it currently stands. While your manuscript cannot be accepted in its present form, we are willing to consider a revised version in which the issues raised by the reviewers have been adequately addressed.  We cannot, of course, promise publication at that time.

In addition to addressing the reviewers' points aimed at clarifying the results, you can more prominently convince readers of the originality of the current model in the context of your previous work.

Sincerely,

Stacey Finley, Ph.D.

Associate Editor

PLOS Computational Biology

Daniel Beard

Deputy Editor

PLOS Computational Biology

[LINK]

Reviewer's Responses to Questions

**Comments to the Authors:**

Reviewer #1: The authors present an expanded version of a model published previously. This version explores the role of two additional positive feedback loops not considered in the old model.

The model analysis and simulations generated several interesting predictions that can be experimentally validated.

Major comments:

1- In the present form, the model is not validated, as the authors did not perform any comparison of model simulations with experimental data. I understand that ideal experimental data to validate the model may not be available, but the authors should, at least, compare PIP2, DAG and PA steady state levels in resting conditions (without chemotactic gradient) with available estimates (even if in different cells) to check that the model is working around a feasible range of concentrations.

2- The authors should also add a sensitivity analysis, so that we can have an idea of the influence of parameter values in model behaviour. The authors have studied the impact of some parameter changes that are more relevant for the studied PFLs, but a overall sensitivity analysis would allow us to identify parameters that should be estimated with higher precision, or detect problems of identifiability. I suggest the authors should compute the relative sensitivity of "Maximum front/back PKC activity ratio" and "dose-response robustness" to small relative changes in each parameter value.

3- The authors have studied a) PFL1 parameter effects and b) geometric effects in the absence of PFL2. When they study PFL2, they identify some parameter conditions where PFL2 does (or does not) contribute to PKC activity polarization. It would be nice to check (even if in supplementary materials only) if the observations made in the absence of PFL2 (both for a) and b) effects) are also seen in both PFL2 scenarios (PFL2 parameters that influence or not PKC activity polarization). If the observations are different, the differences can be a hint to identify the presence and type of PFL2 in a particular cellular system.

4- PA is also known to activate PI4P5K (https://onlinelibrary.wiley.com/doi/full/10.1002/jcb.21027), an enzyme that produces PIP2. So, PA may activate PIP2 synthesis rate. How would this regulatory interaction interfere with the remaining feedback loops in your system?

Minor comments:

1- The meaning of the asterisk (identifying irreversibility of the PKC polarization) should me mentioned in all the figure legends where it appears, and not only in the main text

2- Is it reasonable to have the same diffusivity for DAG and DAG-PKC?

3- In figure 3 C, in the second column, some lines have two numbers (1/16.7). Explain the meaning of the two numbers in the figure legend.

Reviewer #2: In this manuscript, the authors build on two previous model (Refs 17 and 29) to investigate the role of two new feedback loops associated with phosphatidic acid. They show that components of this feedback loops are important regulators of the model and can enhance the robustness of DAG/active PKC polarization. Using this network in a model of wound-healing, they show that efficient collective migration can be achieved with these polarization thresholds. This is an important area of research but the execution of the modeling was lacking.

Major comments (these comments are in the order of my reading the text):

1) Originality: It is very hard to distinguish the substantial new novel contributions from this model as compared to Mohan et al. 2017 [ref 17]. In fact, Figure 1A in very similar in both papers except for the addition of the red reactions in the current manuscript. It is not immediately obvious that the addition of the two new feedback loops as modeled by the reactions in red is a significant advance for publication in Plos Comp Biol. For a reader to understand this manuscript requires a thorough reading of ref 17 making it hard to be a standalone publication.

2) There are many abbreviations and assumptions throughout the text limiting accessibility. For example, in Results paragraph 1, the reasoning underlying how PA affects PLC activity were modeled are really assumptions of the model that must be carefully stated in a separate section. Additionally, these assumptions must be revisited in the limitations section of the discussion. The large number of abbreviations and notational issues also make the manuscript hard to follow. For example, using 'e' for active PLC enzyme is a weird notation. For the longest time, I interpreted to be Euler's number and struggled to follow the terms in the supplementary table. A list of abbreviations would also be helpful, although I'm sure the two feedback loops can be named better than PFL1 and PFL2.

3) Figure 1B: Where is the gradient of the receptors in the spatial geometry? If it is a 2D spatial model (as shown in Figure 3A), then is the receptor gradient simply the difference of receptor at x = 20 and x =-20 divided by 40? Or is it along the arc length? In which case, x doesn't span -20 to +20 but whatever the arc length corresponding to the locations is. Given that it is a 2D spatial model, and the membrane is only the arc length of the curve, the physicality of the imposed receptor gradient doesn't make sense to me.

4) Page 7: This conclusion held true across across simulations with extensive sweeps of the PFL parameters. Where are these shown?

5) Figure 2: The authors should consider using different symbol shapes and not just colors for front and back (this is a minor point). In panel 2A: do the front and back curves overlap throughout? What is most interesting is that the region close to rfrac of 0.1 is where many oscillatory solutions seem to lie. However, these oscillations are not fully explored. Does this oscillation translate into a spatial pattern? These regimes should be carefully analyzed for stability features since these oscillations are observed in simulations and the underlying cause identified. The issue of oscillations was completely glossed over in the manuscript. Could perturbations in rfrac result in loss of steady state or the maximum front/back ratio? Could stochasticity be important (likely yes, but how?)? Addressing these questions would complete the study and significantly elevate the manuscript.

6) Figure 2: Another issue is that membrane density of PKC front to back is compared but what does this mean in a spatial model (see comment regarding Figure 1B, the same issue lies here and throughout the manuscript.)

7) Figure 3 and associated results: This the first place where the geometry is shown and one asymmetry is introduced. The logic for introducing this asymmetry is not clear. But what becomes clear immediately is that shape effects are important. However, this effect is also not fully explored. Given the 2D model (1D membrane) and the role of shape in signaling (see the many references in ref 46, the authors just gloss over this important issue and move on to other things. This significantly diminishes the potential impact of the work. Also, the difference between the two columns in figure 3C is not clear.

8) Figure 4 and 5 and associated results: The authors identify a series of kinetic parameters that are important in determining the ratio of PKC front/back activity. Then they vary these parameters and show that they are important. This is a circular argument. These parameters are important because of the way the model was constructed. Additionally, it is not clear under what physiological conditions these parameters can be measured or manipulated in experiments. Varying copy number of enzymes is a much more meaningful experimental correlate. What are the *s in Figure 4? They are not defined in the caption of this or previous figures.

9) sensitivity analysis: sensitivity analysis of the kinetic parameters and initial conditions should be included, particularly in light of the oscillatory regimes. What if the diffusion constants were altered? Then the results across the entire manuscript would change I would think? It would be important to report to what extent the model is robust to these changes. Plus the diffusion constants must be reported from literature where available.

10) if receptors were allowed to diffuse freely rather than maintain an active gradient across the membrane (after correcting for the geometry issues raised above) would a steady state receptor gradient exist? That seems unlikely given that studies have shown that receptors diffuse in the plane of the membrane and diffusion serves to dissipate the gradient. This should be explored a carefully because this gradient is the input to the model.

11) Testable hypotheses: Regarding the experiments proposed by the authors to test molecular mechanisms, it is not as trivial as they propose to use isoform selective inhibitors. First, the model predicts that kinetic parameters are important, which are near impossible to alter in situ. Interestingly, this section has the least references to plausible experimental modalities, so it is not clear if these methods are even possible.

11) References: the fraction of self citation is rather high. I appreciate that the authors have done a substantial amount of work in this field but they have also missed many fundamental citations in cell shape regulation of signaling (see comment about the references in 46) and receptor diffusion.

Reviewer #3: Please see the attached word document.

**Have all data underlying the figures and results presented in the manuscript been provided?**

Reviewer #1: Yes

Reviewer #2: None

Reviewer #3: Yes

PLOS authors have the option to publish the peer review history of their article (what does this mean?). If published, this will include your full peer review and any attached files.

Reviewer #1: No

Reviewer #2: No

Reviewer #3: No

---

## [Decision Letter · Decision Letter 1]

6 Jan 2020

Dear Dr Haugh,

Thank you very much for submitting your manuscript, 'Mechanistic models of PLC/PKC signaling implicate phosphatidic acid as a key amplifier of chemotactic gradient sensing', to PLOS Computational Biology. As with all papers submitted to the journal, yours was fully evaluated by the PLOS Computational Biology editorial team, and in this case, by independent peer reviewers. The reviewers appreciated the attention to an important topic, and the edits made in the first round of revision. There remains one issue to be addressed - further clarifying what is meant by the use of a linear gradient for the cell surface receptors, particularly in the context of your spatial model.

We would therefore like to ask you to modify the manuscript according to the review recommendations before we can consider your manuscript for acceptance. Your revisions should address the specific points made by each reviewer and we encourage you to respond to particular issues Please note while forming your response, if your article is accepted, you may have the opportunity to make the peer review history publicly available. The record will include editor decision letters (with reviews) and your responses to reviewer comments. If eligible, we will contact you to opt in or out.

- Supporting Information uploaded as separate files, titled 'Dataset', 'Figure', 'Table', 'Text', 'Protocol', 'Audio', or 'Video'.

We hope to receive your revised manuscript within the next 30 days. If you anticipate any delay in its return, we ask that you let us know the expected resubmission date by email at ploscompbiol@plos.org.

Sincerely,

Stacey Finley, Ph.D.

Associate Editor

PLOS Computational Biology

Daniel Beard

Deputy Editor

PLOS Computational Biology

[LINK]

Reviewer's Responses to Questions

**Comments to the Authors:**

Reviewer #1: I am satisfied with the author responses

Reviewer #2: The revised version of this manuscript is significantly improved and I commend the authors for their effort. I only have the following point for clarification.

Figure 1B and Figure 2: I still don't understand what it means to describe a linear gradient of receptors when they are located on the arc length of the membrane. I appreciate the authors clarification from the figure but since this is one of the important features in their model, it should be stated very clearly what this means. I am very confused by this point and I think clarity on this will not only help eliminate confusion but increase accessibility of spatial models.

**Have all data underlying the figures and results presented in the manuscript been provided?**

Reviewer #1: Yes

Reviewer #2: Yes

PLOS authors have the option to publish the peer review history of their article (what does this mean?). If published, this will include your full peer review and any attached files.

Reviewer #1: No

Reviewer #2: No

---

## [Editor Report · Decision Letter 2]

3 Feb 2020

Dear Prof. Haugh,

We are pleased to inform you that your manuscript 'Mechanistic models of PLC/PKC signaling implicate phosphatidic acid as a key amplifier of chemotactic gradient sensing' has been provisionally accepted for publication in PLOS Computational Biology.

Before your manuscript can be formally accepted you will need to complete some formatting changes, which you will receive in a follow up email. A member of our team will be in touch within two working days with a set of requests.

Best regards,

Stacey Finley, Ph.D.

Associate Editor

PLOS Computational Biology

Daniel Beard

Deputy Editor

PLOS Computational Biology

---

## [Editor Report · Acceptance letter]

31 Mar 2020

PCOMPBIOL-D-19-01266R2 

Mechanistic models of PLC/PKC signaling implicate phosphatidic acid as a key amplifier of chemotactic gradient sensing

Dear Dr Haugh,

I am pleased to inform you that your manuscript has been formally accepted for publication in PLOS Computational Biology. Your manuscript is now with our production department and you will be notified of the publication date in due course.

With kind regards,

Laura Mallard
